# Understanding Social Media Literacy: A Systematic Review of the Concept and Its Competences

**DOI:** 10.3390/ijerph19148807

**Published:** 2022-07-20

**Authors:** Karina Polanco-Levicán, Sonia Salvo-Garrido

**Affiliations:** 1Programa de Doctorado en Ciencias Sociales, Universidad de La Frontera, Temuco 4780000, Chile; k.polanco01@ufromail.cl; 2Departamento de Psicología, Universidad Católica de Temuco, Temuco 4780000, Chile; 3Departamento de Matemática y Estadística, Universidad de La Frontera, Temuco 4780000, Chile

**Keywords:** social media literacy, digital literacy, media literacy, systematic literature review

## Abstract

Nowadays, people spend long periods on social media, ignoring the implications this carries in daily life. In this context, the concept of social media literacy, an emerging concept scarcely developed in the literature, is relevant. This study sought to analyze, descriptively, the main definitions and competences of the concept of social media literacy. The methodology included a systematic search of literature in the databases Web of Science, PubMed, and Scopus between 2010 and 2021, applying filters for English and Spanish, including only scientific articles. A total of 1093 articles were obtained. An article selection process took place, applying the inclusion and exclusion criteria, resulting in a total of 15 articles being selected. The findings indicate that the concept of social media literacy is based on media literacy to then integrate the characteristics and the implications of digital platforms. This is linked to the development of cognitive competences, where critical thinking, socio-emotional competences, and technical competences are fundamental, considering the social context. The development of socio-emotional competences stands out since social media are a frequent place of interaction between people.

## 1. Introduction

The transformation of society has been linked to technological changes that are an important part of people’s lives [1,2]. Digital technologies are inserted in aspects of social life, in families and relations with others, at work, in governance and political participation, and they generate new ways to shape a community [3,4,5]. In this sense, social media are widely used by different societies, transcending the geographical borders of territories and cultures, connecting the global to the local [6,7]. Staying on the Internet and social media for extended periods has resulted in media and digital literacy continuing to gain importance [1].

It is important to specify that social media differ from other types of Internet platforms in that they are characterized by their mass use, they allow content creation, and are not only consumed passively, making it possible for people who do not have formal knowledge about mass media to produce information [8]. This is even more relevant considering the cross-sectional use of social media by different age groups and that children’s exposure to cell phone screens begins at an early age [9]. Later, in adolescence they spend extensive periods on social media due to their socializing with their peers [10,11], whereas university students spend an average of 20 h a week on such digital platforms [12], it has been reported that 98.3% of survey respondents state they use social media [13]. The opposite would mean being outside a relevant social space [14]. In the older adult population, there is evidence that they use the technology less other age groups; however, the rates of social Internet use are increasing [15].

It should be noted that users are exposed to different phenomena on social media, such as publicity, images with a positivity bias, and aggressive and violent behaviors. In addition, the way in which social media operate must be considered as they use technology to filter content based on the users’ previous choice, favoring confirmation bias [16]. They also offer the opportunity to choose with whom one wishes to interact, enabling the formation of groups or communities with similar characteristics [17,18,19], which can foster negativity against what is different, which can be particularly relevant in phenomena such as cyberbullying, which has been linked to time spent on social media [20,21].

Thus, there are also messages on social media that can be potentially harmful when they are about health and personal appearance [22], considering people’s exposure to advertising and photos shared with positivity biases [23,24]. In this sense, exposure to photos that have been manipulated to achieve a positive appearance is associated with reducing body image and body satisfaction, with the increase in the desire of young women to get cosmetic surgery [25], depending on the time spent on the Internet [23].

On the other hand, users can be confronted with demands and difficulties such as the dissemination of false and manipulated news in the post-truth era [1,26], which are produced and put into circulation intentionally to obtain benefits such as more visits by users [27]. This is combined with people sharing information without a review process for this content since positive feedback from other users prevails; consequently, fake news goes viral very quickly [26]. People are needed in the role of information consumers; they must develop critical thinking, i.e., a skeptical view of the selection of the news provided through algorithms and the news sources must be tracked [4,26], since discerning veracity or falsity is a responsibility that transcends the individual [5].

It is important to note that the use of social media is not negative in itself as it can increase social capital, foster friendships and reduce feelings of loneliness; however, it depends on the user’s characteristics and how the different platforms are used [28,29]. As a result, teaching and learning competences for the use of these Internet platforms are particularly relevant since they include social and ethical aspects and technical skills [14], as well as competences that can assess information that aids in better decision-making [30].

Media literacy was defined by the Aspen Institute [31] as “the ability to sensitize, analyze and produce information for specific results” (p. 6), although this conceptualization has certainly undergone progressive transformations, moving from printed information to expression and communication that includes new symbolic forms, such as images and multimedia content. In addition, social media have enabled group collaboration and the dialogue of a large number of people who produce content [32]. It is worth noting that Hobbs [32] refers in particular to media literacy and understands it as knowledge, competences, and skills for life that make it possible to participate in today’s society by accessing, analyzing, evaluating, and creating messages in different ways and in different media, being the result of media education. For his part, Buckingham [33] emphasizes the critical component and the understanding that contents are inserted in a broad context, for example, digital capitalism. The emergence of new types of literacy is linked to the appearance of Internet and mobile communication technologies, which have resulted in the appearance of new media. Considering their impact, this is occurring with technologically based sociocultural platforms [34].

In the same vein, digital literacy refers to a broad set of competences around the use of digital media, computers, and information and communication technologies (ITC), being understood as part of other forms of literacy, such as computer, Internet, media, and informational literacy [35]. Currently, efforts are being made by the international community to guarantee digital literacy [36], because since the COVID-19 pandemic time on the Internet and social media has increased [37]. It is important to mention that digital literacy has been proposed as a strategy against social inequality, given the connection between technological exclusion and wider forms of economic and social exclusion [38], because people have fewer opportunities to develop skills due to their limited Internet connection, thereby reducing participation levels [39]. Another relevant element is that it is linked to socio-economic disadvantage with a lack of knowledge about the algorithms that these types of platforms use to recommend content [40].

Literacy in traditional and digital media is central given that we live permanently receiving messages from different sources [41]. Generally, these are focused on improving people’s competences to integrate and operate in today’s society [42]. Therefore, it is necessary to promote the development of skills such as critical thinking because even though teenagers and young adults have known the world with the Internet, they do not have better developed skills in all the areas that digital literacy addresses [43]. Nevertheless, according to Leaning [35], the difficulty arises because media literacy does not sufficiently address digital technology, considering that digital literacy does not fully develop a critical approach compared to media literacy. However, it is relevant to point out that the boundaries between the types of literacy can be blurred; in addition, other proposals progressively emerge that link different approaches such as critical digital literacy, rendering the desired distinctions complex [44,45,46].

In this sense, due to their mass use, social media have transformed the way we relate to each other, form communities, and use mass media. This has been of interest, with proposals on the issue of literacy being generated that focus particularly on these digital platforms. Therefore, Livingstone [47] indicates the need for literacy focused on social media to update the analysis of media literacy. Nevertheless, this concept has limited theoretical development and little operationalization [7,48]. In addition, there is evidence that authors define it differently; it has not been clearly established what the competences are that are included in this type of literacy given the authors working with this concept in their research.

In light of the above, this article focuses on social media literacy by performing a systematic literature review to better understand the concept in terms of the competences it provides that adequately guide efforts in the direction of teaching and learning processes in this area. The relevance of these processes must be borne in mind due to the mass use of such platforms and their use by people of different ages for extended periods, considering there are dangers in social media while at the same time they afford possibilities for interaction, entertainment, and other options that can be useful with an adequate understanding of how social media work and how to make use of them. Therefore, the aim of this study was to analyze, descriptively, the main definitions and competences of the concept of social media literacy.

## 2. Materials and Methods

A systematic search of the literature was done, considering the guidelines of Preferred Reporting Items for Systematic Reviews and Meta-Analyses (PRISMA) [49], in the Web of Science, PubMed, and Scopus databases in July 2021. The question that guided the search strategy was: what are the competences that must be developed to operate on social media? The search took place using free terms and terms from Medical Subject Headings (MeSH) including social media, social media sites, digital literacy, media literacy, and social media literacy. The filters were: language (English and Spanish), number of years (from 2010 to date), and article type (article). With respect to the total articles (*n* = 1039), they were first selected by relevant title, second, by relevant abstract. Then, the articles were reviewed in full (*n* = 59), and the inclusion and exclusion criteria were applied, resulting in 15 articles (Figure 1).

### 2.1. Criteria for Eligibility

Inclusion criteria: Articles were selected that proposed a conceptual definition of social media literacy and/or that demonstrated the competences that integrate this concept. Articles were included where the participants were children, teenagers, young adults, adults, and families. Only scientific articles, theoretical and empirical, in English and Spanish between 2010 and 2021 were included.

Exclusion criteria: Articles that address social media from digital literacy without specifically considering the scope of social media literacy were not included, since they do not define the concept, nor do they refer to the competences that social media literacy encompasses. In addition, articles that address digital platforms but do not consider social media were not included. Theses, conference proceedings, and systematic reviews were not included. Articles in languages other than English or Spanish or with a publication date before 2010 were also excluded.

### 2.2. Procedure

Articles were selected considering the inclusion and exclusion criteria. The articles also had to provide information that responded to the research question; therefore, those articles that did not fit as previously indicated were eliminated. Where questions or disagreements arose about the selected articles, they were resolved through the joint review by the two authors to determine their relevance and to make a decision about their inclusion.

In terms of biases of this study, the language bias was countered by including articles in Spanish and English. In terms of coverage bias, the different databases (Web of Science, Scopus, and PubMed) were reviewed.

### 2.3. Analysis Strategy

With respect to the selection final, the articles were read and reviewed completely, observing if the records provided a conceptual definition of social media literacy or if they reported on the skills that this type of literacy includes. The other criteria of inclusion and exclusion were also considered. The standard quality assessment criteria for evaluating primary research papers were also applied [50].

Later, a table was constructed to present the studies, considering first the authors, type of study, objective, and information on the sample. Then, the main results were transformed in relation to the research question to report on the studies selected and to organize the findings of this study.

In relation to the biases present in articles, generally the records describe full data in their results; moreover, the results were reported according to the analyses used, considering that this is of interest to this review.

## 3. Results

Fifteen articles were obtained for analysis from the following countries: Australia, Belgium, the Czech Republic, Germany, Indonesia, Singapore, Spain, the United Kingdom, and the United States, it being observed that interest in the concept of social media literacy is concentrated mainly in European countries that develop and contribute theoretical and empirical evidence relating to this concept (Table A1 in Appendix A).

### 3.1. Social Media Literacy: Definition

The conceptualization of social media literacy is based on media literacy [51,52,53,54,55,56,57]. However, it is emphasized that social media are oriented to the interpersonal communication that arises from the human need to establish interactions with others [48,52,53]. Thus, according to Vanwynsbergue [56], the focus would be on favoring the efficiency and efficacy of Internet communication, benefitting social relations (Table A1 in Appendix A).

On the other hand, the understanding of the particular characteristics of such platforms is worth noting, in that it is relevant how the information is presented on social media, considering the objectives after posts by both people and advertising, in addition to positivity bias [51,53,54]. Consequently, social media literacy is oriented towards the prevention of risks such as mental and physical health problems [51,53], as well as other types of consequences that can arise from interactions between people, for example cyberbullying, information spreading, and other difficulties [52,53,55].

### 3.2. Social Media Literacy: Competences

With respect to the different competences that encompass social media literacy according to the different studies, there is evidence that cognitive competences appear cross-sectionally in most of the studies searched. These include understanding, analysis, evaluation, synthesis, and the interpretation of the information, added to the assessment of the motive, purpose, realism, and credibility of the publication. Critical thinking is considered fundamental due to the large volume of information to which social media users are exposed [51,53,54,55,56,57,58,59,60,61]. In addition, according to Schreurs and Vandenbosch [54], cognitive competences include a knowledge of traditional media literacy and the dynamics of interpersonal communication on social media (Table A1 in Appendix A).

Similarly, user-generated information requires that they have knowledge of the implications of sharing personal data and the generation of information considering the digital fingerprint, since this information is used by the social media platforms and shared with other companies, so the user must evaluate what content to share [62]. Likewise, Tandoc et al. [63] report on the need to raise awareness about the content recommendation algorithms that transform the social media experience.

The technical or practical competences include the ability to create, review, organize and share contents [57,58], access, find information and use functions such as privacy settings [62], create social media accounts and publish photos and images, and make videos and memes [60,63]. These competences fulfill an important role so people of different ages can perform adequately on these digital platforms [51,55,56,57,58,59,60].

On the other hand, the socio-emotional competences are integrated by several authors into the conceptualization of social media literacy because such digital platforms are oriented to the interaction between different people who share content online; therefore, management strategies for interpersonal communications are relevant [48,51,54,55,56,63]. Festl [48] proposes that the development of social competences is central to social media literacy including participation and moral, communicative, and education competences, consistent with other studies that lend relevance to motivation, attitude, and behavior that people on social media exhibit [55,56]. In addition, Schreurs and Vandenbosch [54] note that effective competences are reflected in the use of adaptive strategies when users are exposed to difficulties on social media, as indicated in Appendix A (Table A1).

The proposals of authors that consider the relevance of the context in which social interactions occur as well as the language used on social media are worthy of note. Specifically, the differences between the different digital platforms must be taken into account since they have particular ways of operating [55]. Moreover, the sociocultural pragmatics in the different social media must be borne in mind, i.e., changes in the users’ language, relations, and behavior depending on the different social and cultural contexts that take place on the Internet [57]. This would make it possible to assess the context that could help discern veracity of the information [60], considering the increase in fake news [63].

## 4. Discussion

The objective of this study was to analyze descriptively the main definitions and competences of the concept of social media literacy. The results yielded 15 studies (Table A1 in Appendix A) that address social media literacy by either conceptualizing it, or by referring to the competences of which it consists. It should be noted that there are studies that, despite using the concept in their articles, do not develop it, or they use it to talk about another type of literacy without making a suitable distinction on the issue [22,64,65].

In relation to the findings of this study, the construction of the concept of social media literacy is based on the knowledge gained through media literacy, to then integrate elements focused on catching the particularities, characteristics, and implications of social media. In this context, it is fundamental to consider the social interactions produced on social media, the possibility of users creating content, the large amount of information that circulates on social media that includes user content and publicity from businesses, as well as the content filtering and recommendation technology. In the same vein, it is suggested that the concept of social media literacy could respond to the requirements of today’s society due to the mass and recurring use of these types of virtual platforms worldwide [47,48,51,52,53,54,55,56].

Consequently, social media literacy is an update of media literacy [47], being oriented to favoring people being able to perform adequately on social media considering the various difficulties that can arise. Without a doubt, the phenomena that occur on social media are not all negative, rather these digital platforms have benefits that could be taken advantage of better if users have greater knowledge and competences [28]. Thus, access to the benefits or opportunities that social media afford, such as the possibility of sharing with friends and relatives, should be promoted, but with strategies to protect against damaging trends or risky behaviors [54].

Generally, the analyzed studies converge in the relevance of cognitive competences in social media literacy. It is worth noting the development of critical thinking because most studies mention it being necessary to obtain a suitable understanding and assessment of the content, being aware of the reliability and credibility of the information [55,56,60], reducing the persuasive influence of mass media through the evaluation of the intention and realism of the content [53,61]. This is not an easy task due to the large volume of information and the anonymity of those who produce the content on social media [57]. In this sense, the knowledge about the algorithms with which social media work acquires relevance, presenting information to the user according to their fingerprint [40].

As Livingstone [52] points out, social media literacy is at the intersection between social and mass media, so that the relevance of socio-emotional competences stands out. The social interactions that take place between users in real time or delay time are one of the characteristics that distinguishes social media from other types of digital platforms or mass media; therefore, different authors have focused on the socio-emotional competences to conceptualize and operationalize the construct [48,51,52,53,54]. In this way, such competences can be considered a protective factor against cybervictimization [66], and a greater prosocial behavior in Internet activities is implied [67], since there are adaptive strategies against negative experiences [54].

With respect to the technical or practical competences, there is evidence that among these are the ability to access, create, review, and share content on social media, adding other functions such as those linked to privacy settings. These competences are considered in a general way; however, social media platforms are different from each other, which is why it is relevant to consider those specific skills that could help people to perform adequately on the different social media. Coincidently, Manca et al. [7] refers to a higher skill level that can be cross-sectional on the different social media and skills specific to each digital platform.

Likewise, studies have shown the relevance of the context in which the content is generated in order to assess its construction [55,57,60]. Then, the specific platform can be considered, the context in which differences in the language used and the forms of interaction between users are reflected. On the other hand, it is important to place social media within a broader social and economic context such as digital capitalism [33], being aware of the objectives of the social media companies such as generating profits [68], transforming the private experience into merchandise [69].

Another finding of this study is the different areas in which studies are being conducted that involve this concept. On the one hand, evidence shows that different authors work with this concept applied to the area of physical and mental health related to body perception [51,53,61,70], developing interventions to reduce eating disorders and the negative impact of exposure to social media because they show idealized appearances, such that social media literacy is considered a protective factor [24,61]. Meanwhile, another group of authors focuses on research with children and adolescents due to the continuous use of social media as a result of their need to establish relations with their peers and how their families mediate the use of digital platforms [48,52,54,58]. Consequently, the development of competences by teenagers is fundamental for them to operate suitably on social media, considering that parents show deficiencies in technical competences and knowledge of social media because they use them less or they use digital platforms passively [54,58].

Finally, the relevance of the analysis and the assessment of news content on social media to determine its veracity stands out in the current context [5,26,60,63]. In this sense, the contribution of social media literacy is significant since it considers aspects of such platforms, because when sharing information, it prioritizes the expectation of positive feedback from other users, or that the content supports one’s personal beliefs and values.

## 5. Conclusions

This systematic review collaborated in the understanding of the construct of social media literacy in its definition and the skills that integrate it, being considered an area of emerging research and that its development is very necessary due to people staying on social media specifically for extended periods. Social media literacy is focused on the development of different abilities that range from the technical to the socio-emotional. In this sense, social media, by making possible and favoring social interactions, bring with them requirements for people to perform adequately on digital platforms, understanding that there is no separation between the digital plane and the physical plane; therefore, a mutual influence is produced that could affect people’s experience by being exposed to the dangers on social media that worsen without the skills to deal with such situations.

On the other hand, the social, economic, cultural, and political context is integrated into the analysis conducted on social media given that such platforms have product advertising, political announcements, and other situations to which social media users are exposed. At the same time, the social media differ from each other, so it is relevant to visualize the characteristics of each of them and their differences, noting they each have their own culture that is reflected in the language, behavior, and interactions generated.

In terms of the limitations of this study, it should be noted that there may be articles that were not detected in the systematic search, or that were not selected for the analysis considering the inclusion and exclusion criteria of this study because the authors used concepts linked to media and digital literacy to refer to the concept of social media literacy. Other databases could be added to verify whether there are new articles and integrate them into the results, contributing to different research questions. Similarly, other types of articles such as systematic reviews or conference proceedings could be added since they were excluded here. With respect to the future lines of investigation, studies must be generated considering the construct of social media literacy and its relation to other constructs such as cyberbullying and cyberaggression as the dangers of social media are considered, making it possible to observe which competences that make up social media literacy are those that would mainly protect against these dangers. In addition, it would be interesting to identify the relations with constructs that reflect if social media literacy facilitate the opportunities that such platforms offer. Finally, other studies could broaden the inclusion criteria by incorporating articles that address social media literacy, although the authors have used other broader concepts or approaches in their research. This way, future studies could analyze and evaluate which of the different literacies that focus on social media obtain the best results.

## Figures and Tables

**Figure 1 ijerph-19-08807-f001:**
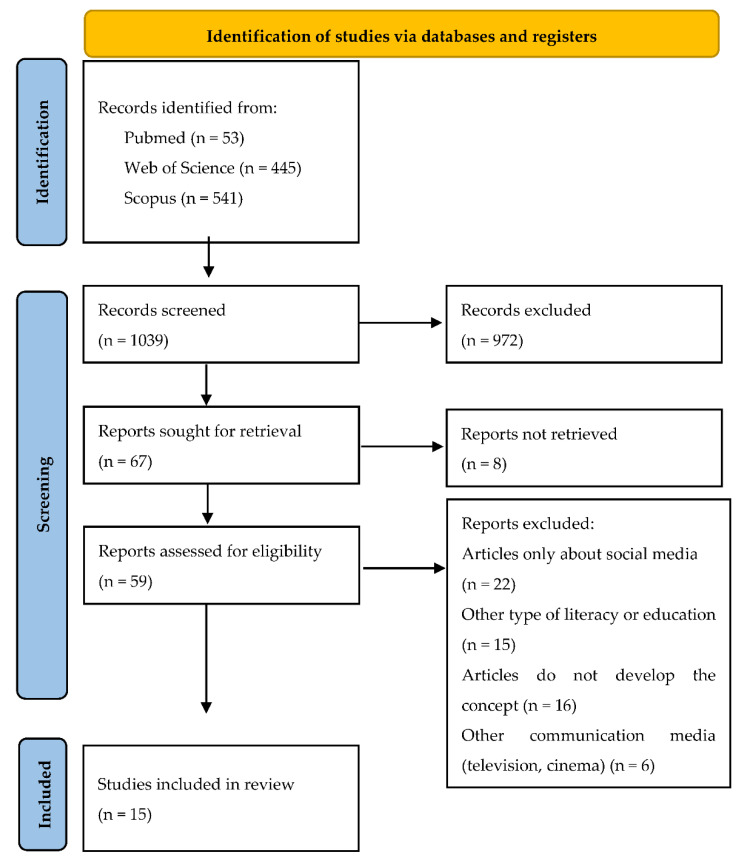
Systematic review flowchart (Adapted from Page et al., 2020 [49]).

## Data Availability

Not applicable.

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
