# Peer review of "Understanding Social Media Literacy: A Systematic Review of the Concept and Its Competences"

_ijerph, 2022, doi:10.3390/ijerph19148807_

Round 1
Reviewer 1 Report
Dear authors,
Your paper is an excellent contribution to defining the concept of social media literacy, considering the previous studies. One step further for future contributions should be to investigate what type of social media literacy could obtain better results.
Author Response
The authors are grateful for the positive assessment of the article.
With respect to the suggestion, this was incorporated into the future lines of investigation (lines 389-391).
The changes made are highlighted in yellow.
Reviewer 2 Report
The manuscript presents a systematic literature review of social media literacy. I suggest accepting the manuscript for publication with minor changes. The following advice may help strengthen your work.
I consider that you are doing a good systematic literature review that sheds light on an interesting topic. However, I see some issues as limitations that you are overseeing. All these issues come from the conceptualization of social media literacy. For instance, you try to establish differences between media literacy, digital literacy, and social media literacy, arguing that media literacy is not digital and digital literacy is not entirely critical (lines 110-111). I disagree with this idea. Despite all these terms referring to different things, the lines between all of them are blurred. Sometimes, when authors refer to media literacy are referring to digital literacies as well. In the same vein, when some authors refer to digital literacy are writing about social media or are presenting a critical digital literacy approach. I’m sure you know the works of Mills (2015) and Massip et al. (2021), clear examples of what I am arguing.
This same issue is related to what you state in lines 278 to 280. In the same way, some articles refer o social media literacy and mean other fields of study, some other articles are writing about critical digital literacy (Castellví et al., 2021) and are writing about social media literacy. I would argue that social media literacy is just a small part of digital literacy or critical digital literacy (depending on the approach). While all social media literacy is digital, not all digital literacy means social media.
Finally, in lines 186 to 188, it seems that you are trying to deal with this issue somehow, but I think you should address it deeper, maybe including some of the references I’m offering you. This is not to say I am asking you to change your research. I consider that narrowing it to the ‘social media literacy’ term is appropriate, but in the limitations section, you should include that there may be other articles referring to this concept that are using broader concepts such as CML. Expanding your systematic literature review to these articles could be interesting for future research.
All in all, I enjoyed reading your article. Thank you for giving me the opportunity to review it.
References
Castellví, J., Tosar, B., & Santisteban, A. (2021). Young people confronting the challenge of reading and interpreting a digital world. Bellaterra Journal of Teaching & Learning Language & Literature, 4(2), e905. https://doi.org/10.5565/rev/jtl3.905
Massip, M., García-Ruiz, C.R., & González-Monfort, N. (2021). Against Hate: hate speeches on digital contexts and alternative counternarratives on Secondary Education students. Bellaterra Journal of Teaching & Learning Language & Literature, 14(2), e909. https://doi.org/10.5565/rev/jtl3.909
Mills, K.A. (2015). Literacy Theories for the Digital Age: Social, Critical, Multimodal, Spatial, Material and Sensory Lenses. Multilingual Matters.
Author Response
The authors are grateful for the positive assessment of the article. In addition, the authors are thankful for the reviewers’ suggestions and observations in the conviction that their contribution has improved the manuscript.
In relation to the changes made in the text:
1. In lines 112 – 115. Information is added about the blurred boundaries between literacies. The references have been incorporated.
2. In lines 372-375 and 385-389. The suggestions made have been added to the limitations of the study and to the future lines of investigation.
The recommendations have been incorporated and are highlighted in yellow.
Reviewer 3 Report
There is a study that is relevant and extremely important for the study of ICT. It is clear the approach and context from which the study is proposed, the referentesteóricos are timely and current. Clear presentation of the methodology, sample selection and relevant elements in the process. It would be key if the authors include the categories of analysis that are used, this gives greater relevance to the results that are then presented.
The results have an important value in what is stated in Table 1, it would be relevant to accompany what is said there with the presentation of specific cases or direct references to the proposed table. Proposed adjustment to the results would give greater coherence to the discussion that is quite strong and leads to various reflections. The conclusions are timely and follow from the above.
Author Response
Thank you very much for the positive evaluation of the article.
In relation to the categories of analysis, it is important to note that, at the time of positing the objectives, the categories of analysis were not initially defined, since it was an inductive process where the categories emerged from the texts and they were not previously defined, considering that is an emerging field of research.
In terms of the results section and its connection to Table 1, the suggestions made were incorporated by adding direct references to the table.
The recommendations have been incorporated and are highlighted in yellow.